# Active Ratio Test (ART) as a Novel Diagnostic for Ovarian Cancer

**DOI:** 10.3390/diagnostics11061048

**Published:** 2021-06-07

**Authors:** Sung-Woog Kang, Adam Rainczuk, Martin K. Oehler, Thomas W. Jobling, Magdalena Plebanski, Andrew N. Stephens

**Affiliations:** 1Hudson Institute of Medical Research, Clayton 3168, Australia; joseph.kang@hudson.org.au (S.-W.K.); adam.rainczuk@hudson.org.au (A.R.); 2Department of Molecular and Translational Sciences, Monash University, Clayton 3168, Australia; 3Bruker Pty Ltd., Preston 3072, Australia; 4Department of Gynaecological Oncology, Royal Adelaide Hospital, Adelaide 5000, Australia; martin.oehler@adelaide.edu.au; 5Robinson Institute, University of Adelaide, Adelaide 5000, Australia; 6Department of Gynaecology Oncology, Monash Medical Centre, Bentleigh East 3165, Australia; tjobling@bigpond.net.au; 7School of Health and Biomedical Sciences, RMIT University, Bundoora 3083, Australia; Magdalena.plebanski@rmit.edu.au

**Keywords:** ovarian cancer, CXCL10, Active Ratio Test, biomarkers, early detection, ELISA

## Abstract

Background: Despite substantial effort, there remains a lack of biomarker-based, clinically relevant testing for the accurate, non-invasive diagnostic or prognostic profiling of epithelial ovarian cancers (EOC). Our previous work demonstrated that whilst the inflammatory marker C-X-C motif chemokine ligand 10 (CXCL10) has prognostic relevance in ovarian cancer, its use is complicated by the presence of multiple, N-terminally modified variants, mediated by several enzymes including Dipeptidyl Peptidase 4 (DPP4). Methods: In this study, we provide the first evidence for the “Active Ratio Test” (ART) as a novel method to measure biologically relevant CXCL10 proteoforms in clinical samples. Results: In a cohort of 275 patients, ART accurately differentiated patients with malignant EOCs from those with benign gynaecological conditions (AUC 0.8617) and significantly out-performed CA125 alone. Moreover, ART combined with the measurement of CA125 and DPP4 significantly increased prognostic performance (AUC 0.9511; sensitivity 90.0%; specificity 91.7%; Cohen’s d > 1) for EOC detection. Conclusion: Our data demonstrate that ART provides a useful method to accurately discriminate between patients with benign versus malignant EOC, and highlights their relevance to ovarian cancer diagnosis. This marker combination may also be applicable in broader screening applications, to identify or discriminate benign from malignant disease in asymptomatic women.

## 1. Introduction

Ovarian cancer is the most lethal of gynaecological cancers, with almost universal recurrence and a 5-year survival rate below 30% [1]. Disease progression is typically asymptomatic, with the majority of patients diagnosed at an advanced stage of progression. Diagnosis prior to extra-ovarian spread is a strong positive prognostic indicator of survival [2]; as such, methods to detect primary and/or pre-malignant lesions are highly desirable in a clinical setting. Despite substantial efforts, however, few biomarker-based tests have been successful in this capacity; currently only the measurement of CA125 and HE4 have demonstrable clinical relevance, and neither is suitable for the detection of early-stage ovarian cancer [3]. Likewise, a combination of CA125 with imaging modalities has also proven ineffective at reducing overall mortality [4]. There is a clear need for improved diagnostic tools to identify low volume, pre-metastatic ovarian cancers prior to spread, and also for accurate monitoring of disease recurrence.

CXCL10 is a member of the CXC chemokine family that mediates the recruitment of CXCR3+ CD4+ Type-1 helper (Th1), CD8+ effector and NK cells to inflammatory sites [5], and has been implicated in the progression of several cancer types. Expression of CXCL10 in breast cancer can enhance tumour-specific T cell infiltration, and is associated with increased overall patient survival [6]; similarly, the delivery of CXCL10-loaded nanoparticles inhibits the growth of liver tumours [7]. Expression of CXCR3 on tumour-infiltrating CD8+ T effector cells also potentiates immune checkpoint inhibition by anti-programmed cell death-1 (PD-L1) in a colorectal cancer model [8]. In ovarian cancers, CXCL10 over-expression defines a subset of high-grade, serous ovarian cancer′s termed “immunoreactive”, highlighting its established functions in adaptive immunity [9]. Induced early during an inflammatory response, CXCL10 (i) promotes leukocyte chemotaxis and (ii) inhibits cell proliferation and promotes apoptosis, dependent on cell type [8]. Despite these well-established roles, CXCL10 expression in ovarian tumours fails to suppress tumour growth as anticipated; over-expression of CXCL10 can lead to enhanced lymph node metastasis [10], whilst decreased plasma CXCL10 concentration has been correlated with positive prognosis [11]. Ovarian tumours also typically exhibit few infiltrating T-effector (Teff) cells, an over-abundance of immunosuppressive T-regulatory cells (Treg) [12], and a homogenous T-cell receptor repertoire substantially different to that found in patient circulation [13]. These findings are at odds with the established role of CXCL10 in promoting inflammatory responses, and have limited its potential prognostic relevance in ovarian cancer to date.

The quantitative measurement of CXCL10 abundance in tumours is further complicated by numerous potential post-translational events, in particular the enzymatic cleavage of its *N*-terminal Val-Pro dipeptide. This truncation is catalysed by dipeptidyl peptidase 4 (DPP4), and results in the conversion of CXCL10 into an antagonist of T cell recruitment that can suppress T-cell migration [14]. In vivo, the activity of DPP4 is strongly correlated with *N*-terminal CXCL10 processing, and directly limits the migration of CXCR3+ lymphocytes to melanoma tumour tissues [15]; accordingly, inhibition of DPP4 enhances tumour suppression in this model through the preservation of biologically active, agonistic CXCL10. Similarly, in patients with chronic hepatitis C virus (HCV) infections, an elevated abundance of cleaved CXCL10 limits the recruitment of circulating monocytes to inflamed liver. Thus, the DPP4-CXCL10 axis has broad biological relevance, and is likely to have prognostic significance in a range of diseases with an inflammatory component [16,17,18].

We previously identified that whilst CXCL10 is abundant in ovarian tumour tissue, it is often present in a truncated form [14]. Post-translational processing of CXCL10 by DPP4 may thus explain observed discrepancies between CXCL10 abundance and poor leukocyte infiltration into ovarian tumour tissues [14]. However, neither the quantitation of this modification nor its prognostic relevance in ovarian cancers have ever been evaluated.

In this study, we report a novel immunoassay termed “Active Ratio Test” (ART) to measure active, full length CXCL10 and its N-terminally cleaved variants. In a retrospective cohort of patients with high-grade, serous ovarian cancer, the ratio of active: total CXCL10 discriminated malignant from non-malignant gynaecological disease samples with greater efficacy than CA125. Further, the combination of active ratio with CA125 and DPP4 achieved robust differentiation of malignant from non-malignant cohorts, as well as successful identification of early (FIGO stage I) from late (FIGO stage III) malignancy. Our data suggest that the active ratio test provides a useful mechanism for preliminary prognostic evaluation of patients with pelvic masses; and could provide a mechanism for the early identification of tumours in women with known genetic predisposition to develop ovarian cancers.

## 2. Materials and Methods

### 2.1. Reagents

Nunc-Immuno Microwell 96-well solid plates were purchased from SigmaAldrich (St. Louis, MO, USA). TMB chromogen solution was from ThermoFisher (Waltham, MA, USA). Recombinant human CXCL10 protein (used for immunization and assay standards) was from Genscript (Piscataway, NJ, USA). Short peptides comprising either the active or truncated N-terminus of CXCL10 were synthesized by Mimotopes (Victoria, Australia). Biotin (Type A) fast conjugation kit, anti-IP-10 antibody, streptavidin peroxidase, anti-human IP-10 ELISA kit, human Dipeptidyl peptidase IV ELISA kit, and active peptidyl arginine deiminase (PAD) cocktail were from Abcam (Cambridge, UK). All other reagents were of analytical grade.

### 2.2. Clinical Samples

Clinical samples were accessed from archival samples stored at −80 °C in the Ovarian Cancer Research Foundation Tissue Bank, collected prospectively from women undergoing surgery for suspected gynaecological malignancies during the period 2007–2014. All samples were obtained from anaesthetised, chemo-naïve patients who had undergone no prior surgical treatment. Histological assessment of tumour type, stage and grade, pre-surgical CA125 measurements, age, menopausal status, pre-existing conditions and any prior history of malignancy were obtained from de-identified patient medical records. Measurement of serum CA125 in the samples was performed in the diagnostic pathology laboratory at the Monash Medical Centre, Melbourne, Australia. Ethical approval (26 June 2002) was obtained from the Southern Health Human Research Ethics Committee (HREC certificates; #06032C, #02031B), with all participants providing prior informed written consent. Sample details relevant to this study can be found in Table 2.

### 2.3. Generation of Monoclonal Antibodies against Human CXCL10

Monoclonal antibodies were generated in the Monash Antibody Technologies Facility against full-length, recombinant CXCL10 protein. Short peptides comprising the intact (NH2-VPLSRTVRCTCISISNQPVNPRSLE-COOH) or truncated (NH2-LSRTVRCTCISISNQPVNPRSLE-COOH) *N*-terminus of human CXCL10, were used for screening. Mice were inoculated intraperitoneally with 16 µg of adjuvanted (Sigma adjuvant system^®^; S6322) full-length CXCL10, co-injected with methylated CpG, in three fortnightly doses. Serum titres were tested by ELISA and compared against naïve sera collected prior to immunization. Mice displaying the highest titres were selected for hybridoma generation. Splenocytes were extracted and fused to SP2/0-Ag14 myeloma cells using polyethylene glycol. The resultant hybridoma cells were grown in azaserine-hypoxantine containing medium in 96-well tissue culture plates for 13 days. The supernatants for each individual hybridoma were screened by microarray for reactivity against both full-length protein and each peptide antigen, with positive clones re-screened by ELISA. The highest responding clones demonstrating appropriate antigen specificity were expanded and sub-cloned, to ensure the derivation of monoclonal hybridoma lines. Monoclonal antibodies were purified from supernatants using Protein G Sepharose, and Ig isotype was determined using commercially available assay kits. The final monoclonal hybridomas lines of interest were grown to 80% confluence, snap frozen with 10% DMSO as cryoprotectant and stored in liquid N2.

### 2.4. Surface Plasmon Resonance Imaging

The binding affinity of monoclonal antibodies mAb-RA2 (detecting only full length, *N*-terminally intact CXCL10) and mAb-RG2 (detecting multiple proteoforms of CXCL10) were analysed by Surface Plasmon Resonance (SPR) using a ProteOn XPR36 SPRi biosensor (BioRad), equipped with a GLC chip. The chip was conditioned with 0.5% SDS, 50 mM NaOH and 100 mM HCl, and then the lanes were activated using equal parts 1-Ethyl-3-(3-dimethylaminopropyl) carbodiimide (EDAC) and N-Hydroxysuccinimide (NHS). Each antibody (50 μg/mL in sodium acetate buffer, pH = 4.5) was immobilised in a separate lane, then deactivated using ethanolamine. Antigens were applied and binding assessed (relative units–RU). All channels were regenerated using 0.85% H3PO3 between applications of antigen. Interspot and control RU were subtracted to give specific binding.

### 2.5. Biotinylation of RA2 and RG2 Monoclonal Antibodies

mAb-RA2 and mAb-RG2 antibodies were biotinylated using a biotinylation kit (Abcam; ab201795) according to the manufacturer’s instructions. In brief, 10 µL of biotin modifier reagent was added to 100 µL of each purified monoclonal antibody (2 mg mL^−1^; PBS, pH = 7.4), followed by gentle mixing. The mixture was added to lyophilised biotin directly and mixed gently. Following 15 min at room temperature, the reaction was quenched by the addition of 10 µL of biotin quencher reagent.

### 2.6. Quantitation of Active and Total CXCL10

Ascites fluid and plasma samples were pre-cleared by centrifugation (18,400× *g*, 20 min at 4 °C) and the cleared supernatants transferred to a fresh tube. Samples were diluted (1:5) with assay buffer (0.1% BSA /0.05% Tween 20 /PBS, pH = 7.4) and maintained on ice prior to assay. CVS samples were vortexed for 30 s, sonicated for 15 min in an ice bath and then vortexed for a further 30 s. Supernatants were centrifuged as above, and maintained on ice until assay.

For assessment by ART, anti-human IP-10 polyclonal antibody (Abcam; #ab9807) prepared in the coating buffer (15 mM NaCO_3_ and 35 mM NaHCO_3_, pH = 9.6) was immobilised on the 96-well microplate (100 µL per well) at a coating concentration of 0.5 µg mL^−1^ for 2 h at room temperature. Plates were washed once with 280 µL of wash solution (0.05 % Tween 20/Milli-Q H_2_O), then incubated with 300 µL blocking solution (5% BSA/0.05% Tween 20/PBS, pH = 7.4) for 2 h at room temperature. Plates were washed four times in wash solution, and then standards (recombinant CXCL10, 48.8 pg mL^−1^ to 200,000 pg mL^−1^) or samples were added to respective wells (100 µL per well) in quadruplicate. Following 2 h incubation, each well was washed four times in wash solution. Biotinylated detection antibodies mAb-RA2 and mAb-RG2 (1 µg mL^−1^ in assay buffer) were added (100 µL per well) as appropriate and incubated for 1 h at room temperature. Following incubation, plates were washed five times with wash buffer, and then incubated with diluted (1:1000) streptavidin peroxidase (1 µg/mL/well) for 45 min at room temperature. Plates were washed a further four times in wash solution, and then 100 µL of TMB chromogen solution added to each well and incubated for 20 min in the dark at room temperature. The reaction was stopped by addition of stop solution (1M hydrochloric acid, 50 µL per well). Absorbance was measured at a wavelength of 450 nm, using a Cytation 3 multimode plate reader (Biotek, La Jolla, CA, USA) equipped with Gen5 v3.08 analysis software. Comparison between ART and standard ELISA (detecting total CXCL10 only) was performed using a colorimetric anti-human IP-10 ELISA kit (Abcam #ab100579) according to the manufacturer’s instructions.

### 2.7. Statistical Analyses

All statistical analyses were performed using GraphPad PRISM (GraphPad Software, La Jolla, CA, USA). All assay data was log-transformed to approximate normality. The best-fit line was determined by nonlinear regression curve in asymmetric sigmoidal (5PL) against recombinant CXCL10 concentrations to quantitate active CXCL10 (mAb-RA2) and total (mAb-RG2) CXCL10, respectively. Significance was determined using one-way ANOVA and pairwise Student’s t-test. For groups with significantly different variance, Welch’s correction was applied. Spearman rank test was used for correlation analyses. Cohen’s d was determined by calculating the mean difference between the two patient groups and divided by the pooled standard deviation. Results of p<0.05 were considered significant.

## 3. Results

### 3.1. Generation of a Functional Assay Differentiating Active from Total CXCL10

We first developed monoclonal antibodies to differentiate between the *N*-terminally intact, chemotactically active form of CXCL10 and all other variants. Using full-length CXCL10 protein as the antigen, hybridoma clones were isolated and screened for reactivity against (i) full-length CXCL10 protein; and (ii) short synthetic peptides, representing either the intact or truncated *N*-terminus of CXCL10. Antibodies secreted by clone mAb-RA2 recognized both full-length and *N*-terminally intact CXCL10, but not the *N*-terminally truncated form (Table 1). Antibodies secreted by clone mAb-RG2 reacted with both proteoforms of CXCL10 tested. SPR analysis demonstrated binding affinity in the nM to pM range in each case.

Using biotinylated mAb-RA2 and mAb-RG2 as detection antibodies, we constructed sandwich ELISAs to independently measure intact or total CXCL10. Specific parameters including limit of detection (LOD), limit of quantitation (LOQ), linearity, inter- and intra-assay precision and dynamic range were assessed (Table 1). Each of mAb-RA2 and mAb-RG2 demonstrated good affinity for full-length CXCL10, with an R-squared of 0.996 and 5 orders of magnitude (OOM) dynamic range (Figure 1). Limits of detection for mAb-RA2 and mAb-RG2 were 95.9 pg mL^−1^ and 116.3 pg mL^−1^, respectively (Table 1). Inter-assay precision was assessed using 10 independently prepared replicates at a single dose; and intra-assay variation was determined between 10 separate assay runs performed on different days. Coefficient of variation (CV) for both intra- and inter-assays was below 10% in each case, demonstrating good reproducibility and assay precision (Table 1).

We next sought to validate quantitative detection of CXCL10 using ART, compared with a commercially available ELISA kit. CXCL10 is abundant in ascites fluid from ovarian cancer patients, where it is involved in the CXCR3-mediated migration of cancer cells and specific T-cell subsets [5,19]. We therefore assessed CXCL10 detection in ascites fluid from ovarian cancer patients (*n* = 212) as a representative, complex biological matrix. Whilst there was considerable deviation in concentration range within the sample group (CVs ranging from 88.4–225.1%), there was no significant difference in quantitation of total CXCL10 between ART versus commercial ELISA (Figure 2A). Positive correlations in quantitated total CXCL10 were observed between ART and commercial ELISA for both benign and malignant ascites fluid, resulting in r values of 0.3084 and 0.2594, respectively (Figure 2B,C).

### 3.2. mAb-RA2 and mAb-RG2 Differentiate Functional from Non-Functional CXCL10

In addition to proteolytic *N*-terminal truncation of CXCL10, other post-translational modifications can influence CXCL10 activity [20]. In particular, deamination of arginine to citrulline (R5Cit) in the *N*-terminal region of CXCL10 is an established functional modification occurring in vivo, and thus must be accounted for in any assay to assess CXCL10 function [21]. We therefore investigated whether the R5Cit modification may influence the detection of full-length CXCL10 by either mAb-RA2 or mAb-RG2.

Recombinant, full-length CXCL10 was incubated with protein-arginine deiminase 2 (PAD2) to induce deamination at R5, and proteins were separated by SDS PAGE and analysed by Western blotting. In vitro citrullination had no effect on CXCL10 detection using mAb-RG2 (detecting total CXCL10), suggesting that mAb-RG2 bound to CXCL10 regardless of the post-translational modification (Figure 3A). By contrast, mAb-RA2 showed substantially reduced detection of R5Cit-CXCL10 consistent with modification of the epitope following R5Cit modification at the *N*-terminus.

A similar result was obtained when R5Cit-CXCL10 was evaluated by ELISA. Citrullination substantially reduced binding of mAb-RA2, whilst mAb-RG2 remained to be able to detect R5Cit-CXCL10 (Figure 3B). Together, the data demonstrate that mAb-RA2 and mAb-RG2 can be used effectively to differentiate active from total CXCL10.

### 3.3. Active Ratio Test Differentiates Benign from Malignant Ovarian Cancer Samples

Ovarian cancers are characterised by poor T-cell infiltrate, and in particular the ratio of T-regulatory: T-effector cells (Treg:Teff) is an important prognostic indicator associated with outcome. As a key agonist of T-cell chemotaxis, over-expression of CXCL10 in high-grade, serous tumours should promote Teff recruitment and improved T-cell infiltrates [5,14]. However, whilst elevated CXCL10 expression consistent with an “immunoreactive” molecular signature is associated with improved prognosis, plasma CXCL10 concentration counter-intuitively displays an inverse relationship with positive prognosis [14]. We previously identified that malignant ovarian tumour tissues contained cleaved CXCL10, which correlated with reduced T-cell infiltrate [14]. We therefore hypothesized that the relative measurement of functional CXCL10, rather than total CXCL10, may be required to properly assess differences between benign and malignant disease.

To establish proof-of-principle for the use of ART to discriminate between benign vs malignant disease, we measured active vs total CXCL10 concentrations in ascitic fluid harvested from patients with either benign or malignant ovarian tumours (Table 2). Consistent with prior studies [14,22], total CXCL10 was elevated in malignant (853.1 ± 1574.0 pg mL^−1^) compared with benign (160.8 ± 362.0 pg mL^−1^) ascites fluid (Figure 4A). Similarly, a broad range in active CXCL10 measurement was also observed (240.4 ± 410.5 pg mL^−1^ and 818.6 ± 1098.0 pg mL^−1^, respectively) (Figure 4B). This broad deviation within the population (CV% ranging between 134.1 and 225.1%) results in low sensitivity for differentiation between benign and malignant samples (Table 3), and highlights the difficulties associated with the direct CXCL10 measurement for diagnostic or prognostic purposes.

Since proportionately greater levels of active CXCL10 were observed in benign vs malignant samples, we examined the ratio of functional: total CXCL10 in each sample, as a mechanism to normalize CXCL10 function between patients (Figure 4C). This “active ratio” measurement significantly improved separation between the groups, resulting in differentiation between benign and malignant samples with cut-off values of <1.43 and <1.25 at 90% and 95% specificities, respectively (Table 3). This relationship also remained valid when malignant samples were separated according to disease stage (FIGO stage I vs stage III; Figure 4D). Whilst plasma CA125 was significantly elevated in patient with malignant vs benign disease (Figure 5A), there was no apparent correlation between active ratio measured in ascites fluid and plasma CA125 levels (Figure 5B). Thus, active ratio provides a marker of malignancy independent of plasma CA125 concentration.

### 3.4. DPP4 Abundance and Activity Do Not Correlate with Functional CXCL10

DPP4 catalyses the removal of a Val-Pro or Ala-Pro dipeptide form the *N*-terminus of CXCL10, converting it into an antagonist of T-cell recruitment. Previous work therefore sought to quantify the level of DPP4-cleaved CXCL10 in biological fluids as an indicator of disease [18]. However, multiple modifications to CXCL10—both proteolytic and non-proteolytic—can result in an expanded repertoire of CXCL10 variants beyond simple DPP4-catalysed *N*-terminal processing [20,21]. Failure to account for these multiple variants thus leads to discrepancies between the measured vs total amounts of CXCL10 in clinical samples and does not adequately capture the functional status of CXCL10 [18].

To clarify the relationship between DPP4 activity and CXCL10 function in clinical samples, we measured DPP4 abundance and specific activity in the same set of benign and malignant ascites. There was no significant difference in abundance or activity observed in either case (Figure 6A,B). Whilst DPP4 abundance correlated (*p* = 0.002) with the calculated active ratio in malignant samples, its specific activity did not; interestingly, benign samples displayed an inverse (non-significant) trend suggesting a correlation between specific activity and active ratio measurement (Figure 6B). These data suggest that whilst DPP4 activity may be related to CXCL10 function in non-malignant disease, multiple CXCL10 variants are likely to be important in determining the overall functional status of CXCL10 in malignancy.

### 3.5. Active Ratio Provides Prognostic Discrimination between Benign and Malignant Disease

To evaluate whether the active ratio could provide useful clinical information, we assessed its performance in ascites fluid using receiver operating characteristic (ROC) curves (Figure 7). Active ratio achieved higher AUC and a substantial improvement in sensitivity/specificity compared with measurement of either active or total CXCL10 alone (Figure 7A, Table 3), highlighting its enhanced utility over these single measurements. Sensitivity, specificity and predictive values of active ratio were also higher than plasma CA125 (Figure 7B; Table 3). Active ratio also achieved greater effect size (Cohen’s d) than either DPP4, CA125 or CXCL10 (active or total) measurements alone (Table 4). Accordingly, the combination of active ratio, DPP4 and plasma CA125 measurement yielded an AUC > 0.95 with good PPV (87–94%) and NPV (93–95%) for the discrimination between benign and malignant disease. Thus, the combination of ART with DPP4 and plasma CA125 proved to be useful in the discrimination between benign and malignant disease in patients presenting with ascites fluid.

### 3.6. Potential Application of ART as a Clinical Diagnostic

Whilst plasma is a minimally invasive sample of choice for biomarker-based testing, the measurement of systemic CXCL10 concentrations could be confounded in certain scenarios (for example, those with an inflammatory component such as rheumatoid arthritis [23]. We therefore explored whether the specificity of active ratio measurements could be enhanced using cervicovaginal swabs (CVS) as a less complex sample type, directly representative of the reproductive tract.

In parallel to our findings in ascitic fluid, active ratio measurement performed on CVS extracts (*n* = 50/group) displayed good discrimination between benign and malignant disease samples with an AUC of 0.80 (*p* < 0.0001) (Figure 8A, Table 4). In the matched plasma samples, ART was also able to discriminate between groups (AUC 0.78, *p* = 0.0002). To explore diagnostic utility beyond ROC analyses, we also examined effect size (Cohen’s d) for each biomarker and sample type (Table 4). Measurement of plasma CA125 returned an AUC of 0.83, second only to ART measurement in ascites fluid (AUC 0.86); however, the effect size was “medium” (Cohen’s d = 0.62). By contrast, ART measurements achieved “large” effect sizes in ascites fluid and CVS (Cohen’s d = 0.86 and 1.0, respectively) suggesting greatly reduced deviation within the same population. Similarly, ART measurement performed in plasma returned a Cohen’s d = 0.79 (Table 4). This result suggests that active ratio is the preferred biomarker to achieve statistically robust measurements, and that its analysis using CVS can provide a robust and clinically useful measurement to discriminate between malignant and non-malignant disease.

## 4. Discussion

Ovarian cancers typically progress in the absence of symptoms, resulting in a majority of advanced-stage diagnoses. In addition, discrimination between benign and malignant pelvic masses is not possible using current imaging techniques; and CA125, the current “gold standard” biomarker, is largely restricted to monitoring tumour recurrence. There remains an unmet need for new biomarker-based assays to improve detection and prognostic profiling of ovarian cancer patients.

Whilst evidence suggests that CXCL10 plays an important role in tumour pathogenesis, its role in ovarian cancers has remained unclear. CXCL10 is secreted by cancer cells in vitro in response to inflammatory stimuli [5]; and in “immunoreactive” ovarian tumours, with increased relative expression of CXCL10 and CXCR3, there is increased recruitment of T-lymphocytes and subsequent positive impact on survival [24,25,26]. However, increased CXCL10 abundance has also been correlated with enhanced lymph node metastasis in other models [10,27,28], whilst decreased concentrations of CXCL10 in plasma have been associated with improved progression-free and overall survival for ovarian cancer patients [11]. Combined with our previous work demonstrating the presence of cleaved CXCL10 in tumour tissue [14], our data strongly indicate that these discrepancies relate to the relative levels of biologically active versus modified CXCL10 associated with ovarian tumours. To provide a biologically relevant assessment of CXCL10 contribution to tumour pathogenesis, both the active and inactive forms of CXCL10 must therefore be accounted for.

Prior study assessed the contribution of different CXCL10 proteoforms to HCV as a potential diagnostic or prognostic tool [18]. However, limited antibody sensitivity and reliance on the detection of the DPP4-catalysed, dipeptide truncation on CXCL10 did not account for all modified proteoforms in the sample, resulting in failure to quantitate cleaved CXCL10 appropriately [16]. Multiple proteolytic modifications to CXCL10, beyond initial dipeptide processing by DPP4, can occur to influence both function and receptor binding [29,30]. In addition to proteolysis, CXCL10 may also undergo citrullination by protein-arginine deiminase type-2 (PAD2) [20,21], further complicating quantification of its biologically active forms. Of particular importance, our mAb-RA2 antibody—used to detect biologically active (i.e., agonistic) CXCL10—did not react with the citrullinated (antagonistic) variant; whilst mAb-RG2 remained to be able to quantitatively detect citrullinated CXCL10 amongst the total species. Moreover, our assay correlated well with a commercial ELISA designed to detect only total CXCL10. Our assay therefore provides an accurate quantitation of chemotactically active CXCL10 against a range of proteolytic and otherwise post-translationally modified variants, from which biological relevance can be appropriately inferred.

Standardisation of collection and handling procedures is essential for biomarker-based assays, where results can be affected by variables including sample timing, extraction methodologies and operator error. Rather than absolute quantitation, we hypothesized that the ratio of active: inactive CXCL10 would provide a robust and consistent measurement, independent of external influences such as sample size or volume. This ratio out-performed CA125 for the discrimination of malignant versus benign disease, with substantially reduced %CV and increased effect size. Effect size was specifically assessed because no two sample means are identical, making dependence on a sole p-value unreliable [31].

As the clinical gold standard, biomarker combinations with CA125 can provide additive diagnostic efficacy and increased sensitivity and specificity compared with single biomarker assays [32,33]. We therefore assessed the combination of active ratio with CA125 and DPP4. Discriminatory power became even more pronounced using this combination, which achieved 93.4% and 94.7% PPV and NPV, respectively, and an overall AUC of 0.95. This data demonstrates that the active ratio can provide superior primary discriminative power for detection of ovarian cancer compared with the current CA125 standard.

An interesting finding in this study was the lack of discriminatory capacity offered by measurement of circulating DPP4. The regulation and function of DPP4 in cancers is complex; DPP4 abundance is reduced in some cancers including gastric [34], but overexpressed in others including ovarian [14]; and its abundance and activity do not correlate with extent of metabolic inflammation [35]. This may be due to the fact that DDP4 has a number of different biological activities reflecting its diverse functions in the malignant tumour environment [36]. In addition, circulating DPP4 can derive from multiple biological sources, and these may contribute to different, non-overlapping biological roles [37,38,39]. In ascites fluid, we observed that DPP4 abundance was negatively correlated with active ratio in malignant disease, but that its specific activity correlated only in benign disease. This suggests that whilst DPP4 activity is a key determinant of CXCL10 function in benign conditions, additional proteolytic processing of CXCL10—for example by matrix metalloproteinases (MMPs)—is likely to contribute both to CXCL10 processing and the regulation of DPP4 activity itself [40]. This further highlights the importance of our approach, measuring active CXCL10 against its inactivated forms.

This study provides the first evidence that an assay to measure biologically relevant CXCL10 proteoforms can be used to discriminate between benign and malignant EOCs—independent of stage—with high efficacy in a retrospective cohort. As a minimally invasive approach, cervicovaginal swab sampling and ART analysis may improve the detection of early-stage ovarian cancers in asymptomatic women, or to monitor relapse or chemo-response.

## 5. Conclusions

Our data provide the first evidence for the Active Ratio Test (ART) as a novel approach to the detection of ovarian cancers. ART provided excellent discrimination between benign and malignant ovarian tumours, and was equally efficient for the detection of early or late-stage disease. The combination of ART with CA125 and DPP4 measurement enhanced efficacy for the differentiation of malignant disease in the cohort. Moreover, ART provided good discrimination with large effect size when evaluated using cervicovaginal swabs, a minimally invasive, clinically established procedure that can be easily standardized for regular use. Increased cohort numbers are now required to expand this data set and provide further validation, prior to formal clinical trials.

## 6. Patents

Aspects of this study are detailed in 530866PCT.

## Figures and Tables

**Figure 1 diagnostics-11-01048-f001:**
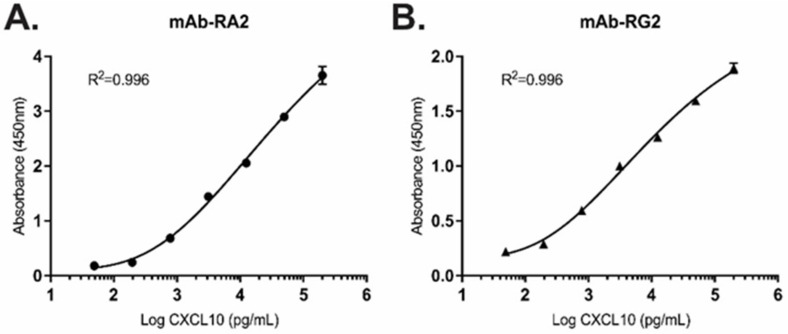
Representative standard curves for the detection of recombinant, full-length CXCL10 using (**A**) mAb-RA2 and (**B**) mAb-RG2 in the range 49–200,000 pg mL^−1^. Mean +/− SD; triplicate measurements per data point.

**Figure 2 diagnostics-11-01048-f002:**
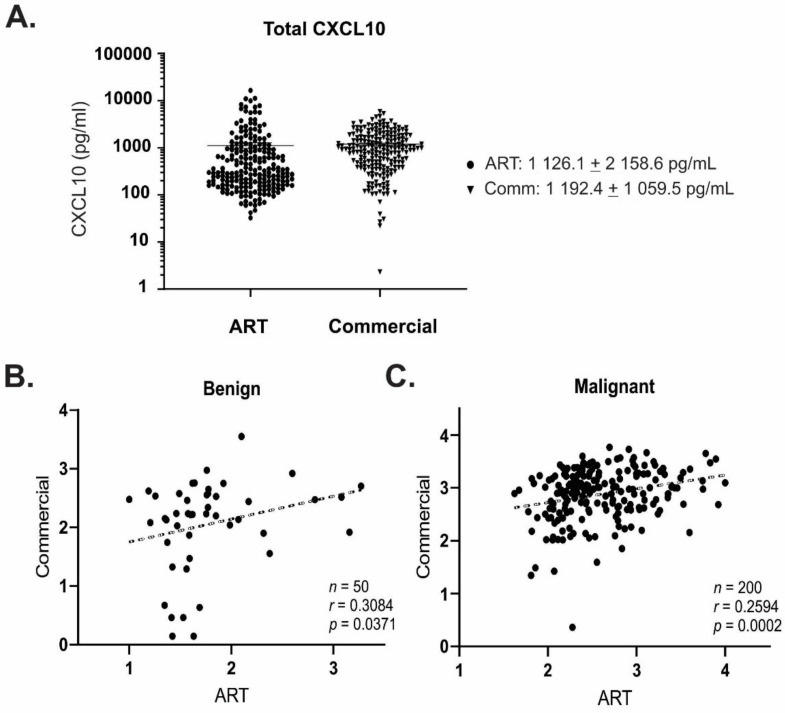
(**A**) Comparison of total CXCL10 detection using ART or commercial ELISA in ascites fluid from ovarian cancer patients (*n* = 212). Data is mean +/− SD, triplicate replicates per sample; (**B**,**C**) Significant correlations were observed between measurements made using ART or commercial ELISA for both (**B**) benign (*n* = 50; *p* = 0.0371); and (**C**) malignant (*n* = 200; *p* = 0.0002) sample types.

**Figure 3 diagnostics-11-01048-f003:**
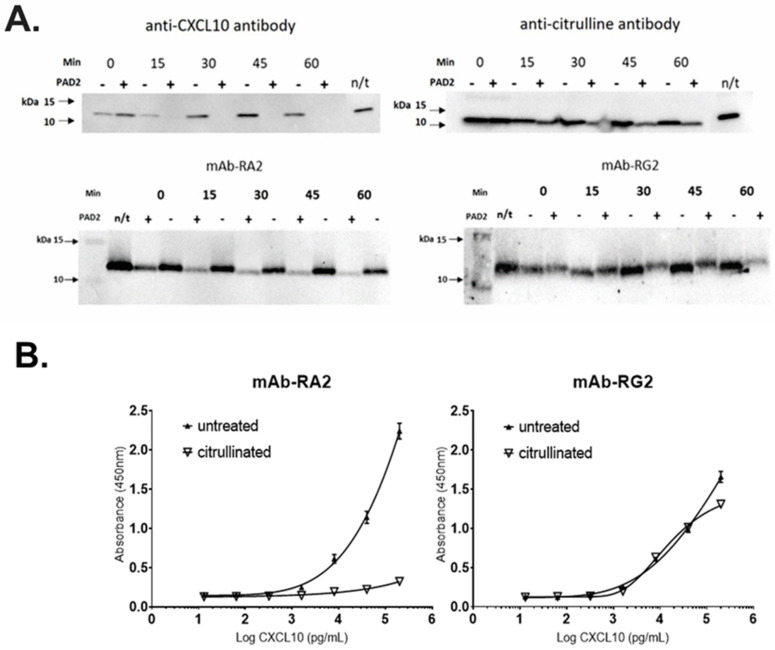
ART successfully discriminated citrullinated from non-citrullinated CXCL10. (**A**) In vitro citrullination of recombinant, full length CXCL10 by PAD2 resulted in loss of detection by commercial anti-CXCL10 antibody ab9807 (upper left), and a substantial reduction in detection by mAb-A2 (lower left). Citrullination was confirmed using an anti-citrulline antibody (upper right). Detection of citrullinated CXCL10 by mAb-RG2 was unaffected. (**B**) ELISA using mAb-RA2 (left) or mAb-RG2 (right) confirmed that detection of citrullinated CXCL10 by mAb-RA2 was abrogated, whilst detection by mAb-RG2 was unaffected. *n* = 3 replicates per data point.

**Figure 4 diagnostics-11-01048-f004:**
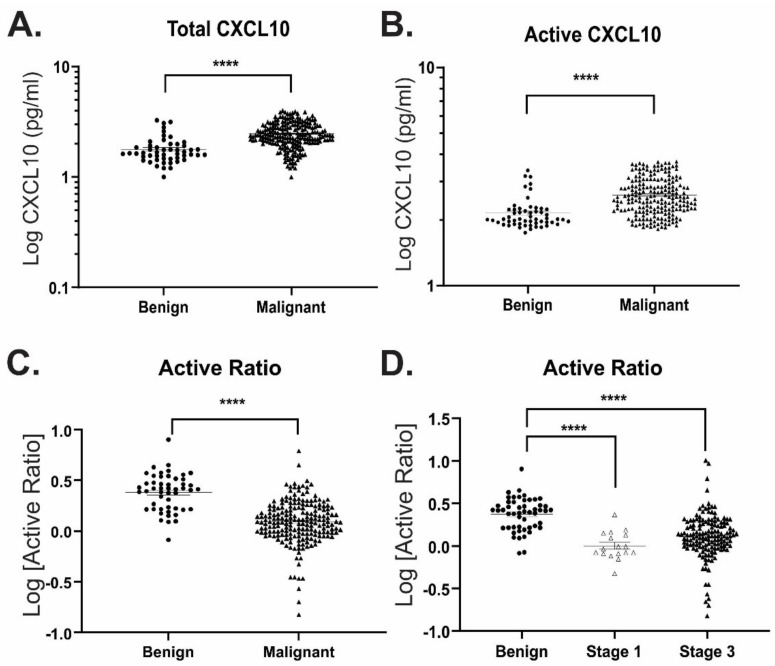
ART differentiates between benign and malignant disease in ascites fluid from ovarian cancer patients. (**A**) Total and (**B**) active CXCL10 concentrations measured in benign (*n* = 51) and malignant (*n* = 212) ascites fluid samples. (**C**,**D**) Calculated active ratio between (**C**) benign and all malignant samples; or (**D**) benign and cancer samples separated according to FIGO disease stage. **** *p* ≤ 0.0001.

**Figure 5 diagnostics-11-01048-f005:**
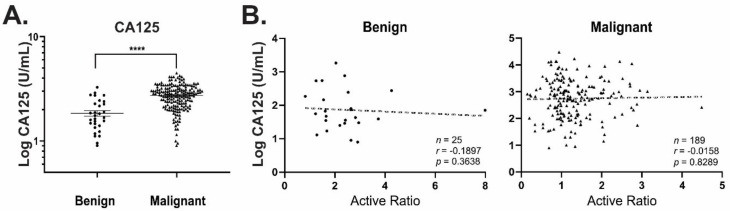
CA125 does not correlate with active ratio. (**A**) Plasma CA125 was measured for patients with either benign (200.8 ± 368.7 U mL^−1^; *n* = 48) or malignant ovarian tumours (1697.0 ± 3409.0 U mL^−1^; *n* = 188). Mean +/− SD, **** *p * ≤ 0.0001. (**B**) No significant correlation was observed between CA125 and active ratio in either case.

**Figure 6 diagnostics-11-01048-f006:**
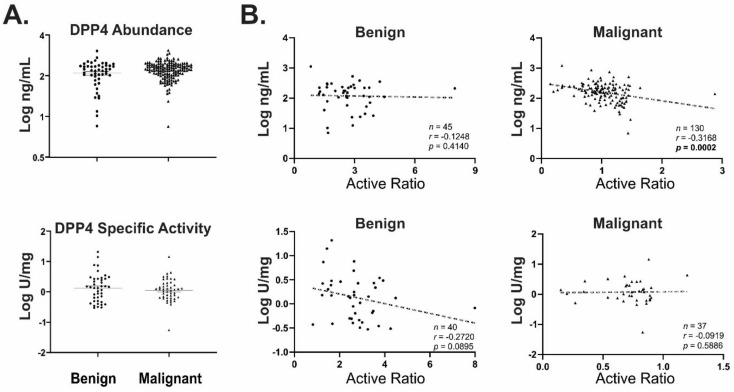
DPP4 abundance and specific activity in clinical samples. (**A**) DPP4 abundance (benign *n* = 49; malignant *n* = 148) and specific activity (benign *n* = 49; malignant *n* = 50) in patient’s ascites fluid were measured by ELISA or activity assay, respectively. (**B**) Correlations between active ratio and DPP4 abundance (upper panel) or specific activity (lower panel) in matching ascites samples.

**Figure 7 diagnostics-11-01048-f007:**
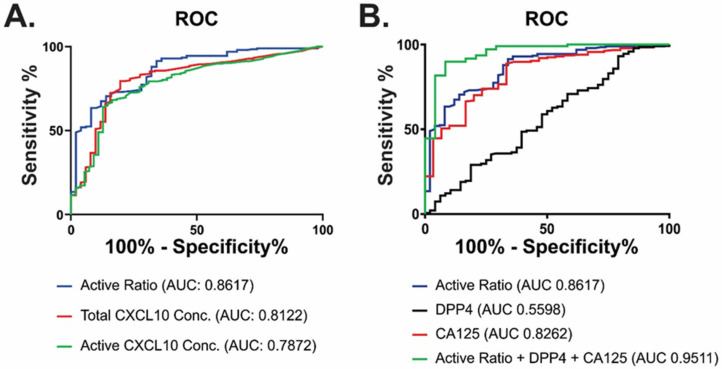
Marker combinations using ART discriminate between benign and malignant disease. Receiver operator curves (ROC) were used to evaluate the discriminatory power of ART alone compared with (**A**) individual measurements of total or active CXCL10; or (**B**) CA125 alone, DPP4 alone or a combination of CA125, DPP4 and ART in ascites samples.

**Figure 8 diagnostics-11-01048-f008:**
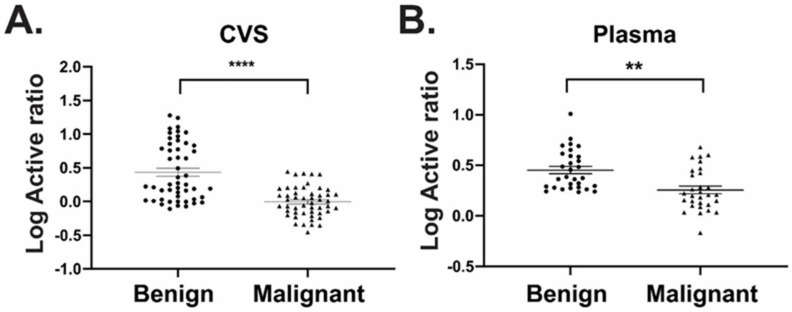
ART performed using either plasma or cervicovaginal swabs discriminated between benign and malignant disease. CXCL10 was measured using ART in (**A**) cervicovaginal swabs (CVS) and (**B**) matched plasma samples from patients with either benign or malignant ovarian cancer. *n* = 50/group; **** *p* < 0.0001, ** *p* < 0.01.

**Table 1 diagnostics-11-01048-t001:** Binding affinity and precision of mAb-RA2 and mAB-RG2.

	Binding Affinity (KD) for CXCL10 Proteomics			Assay Parameters			
	Full Length	Intact *N*-term	VP-Truncated *N*-term	Limit of Detection (LOD)	Limit of Quantification (LOQ)	Intra-Assay Precision (%)	Inter-Assay Precision (%)
**mAb-RA2**	4.90 × 10^−10^	7.36 × 10^−10^	no binding	95.9 pg mL^−1^	162.2 pg mL^−1^	2.8	9.9
**mAb-RG2**	3.89 × 10^−12^	3.54 × 10^−9^	3.00 × 10^−9^	116.3 pg mL^−1^	272.7 pg mL^−1^	4.0	8.5

**Table 2 diagnostics-11-01048-t002:** Summary of patient samples (ascites, plasma, CVS) used for comparison of active CXCL10 ratios between benign and malignant samples *.

Group	Pathology	Grade	Stage	Median CA125 (IQR)	Menopausal Status
Benign (*n* = 51)	Cystadenoma (*n* = 19) Fibroma (*n* = 10) Other (*n* = 22)	n/a	n/a	52 (23–154)	Mixed
Malignant (*n* = 224)	Serous (*n* = 163) Mucinous (*n* = 10) Endometroid (*n* = 14) Clear cell (13) Mixed epithelial/poorly differentiated (*n* = 24)	1–3	I–IV	641 (149–1591)	Mixed

* See Appendix A for individual details.

**Table 3 diagnostics-11-01048-t003:** Prognostic performance of active ratio and other biomarkers in the ascites fluid from ovarian cancer patients to distinguish between benign and malignant.

Biomarkers (Cut-Off Points)	Specificity % (95% CI)	Sensitivity % (95% CI)	AUC	PPV	NPV
**Active Ratio**					
<1.43	90.0 (78.6–95.7)	63.5 (56.6–69.9)	0.8617	80.4	79.2
<1.25	96.0 (86.5–99.3)	52.0 (45.1–58.8)		89.4	75.5
**Total CXCL10 (pg mL^−1^)**					
>241	90.2 (79.0–95.7)	51.0 (44.2–57.6)	0.8122	77.1	74.0
>1204	94.1 (84.1–98.4)	19.5 (14.7–25.4)		68.2	64.4
**Active CXCL10 (pg mL^−1^)**					
>598	90.1 (80.4–96.1)	35.6 (29.4–42.3)	0.7872	70.0	68.4
>1400	94.6 (85.2–98.5)	17.3 (12.8–23.0)		67.5	63.9
**DPP4 (ng mL^−1^)**					
>316	89.6 (77.8–95.5)	12.2 (7.8–18.4)	0.5598	43.2	61.2
>389	95.8 (86.0–99.3)	7.4 (4.2–12.8)		53.3	61.5
**Plasma CA125 (U mL^−1^)**					
>548	90.0 (73.5–97.9)	52.1 (44.7–59.5)	0.8262	77.1	74.4
>757	93.3 (77.9–99.2)	44.7 (37.4–52.1)		89.8	73.0
**Combined Biomarkers**					
**Active ratio & DPP4** (<1.43 and > 316 ng mL^−1^)	91.7 (73.0–99.0)	77.3 (68.3–84.7)	0.9114	85.8	86.2
**Active ratio & DPP4** (<1.25 and > 389 ng mL^−1^)	95.8 (78.9–99.9)	67.3 (57.7–75.9)		91.2	81.9
**Active ratio & plasma CA125** (<1.43 and > 548 U mL^−1^)	91.7 (70.3–86.3)	79.1 (70.3–86.3)	0.9337	86.1	87.1
**Active ratio & plasma CA125** (<1.25 and > 757 U mL^−1^)	95.8 (78.9–99.9)	51.8 (42.1–61.5)		88.9	75.4
**Active ratio & DPP4 & plasma CA125** (<1.43 and > 316 ng mL^−1^ and > 548 U mL^−1^)	91.7 (73.0–99.0)	90.0 (82.8–94.9)	0.9511	87.5	93.4
**Active ratio & DPP4 & plasma CA125** (<1.25 and > 389 ng mL^−1^ and > 757 U mL^−1^)	95.8 (78.9–99.9)	81.8 (73.3–88.5)		93.4	94.7

**Table 4 diagnostics-11-01048-t004:** Effect sizes of biomarkers based on sample types.

Biomarker	Sample Type	AUC	Cohen’s d	CV % (Malignant)	*p*-Value
Total CXCL10 (pg mL^−1^)	Ascites	0.8122	0.6976	191.7	<0.0001
Active CXCL10 (pg mL^−1^)	Ascites	0.7872	0.6062	134.1	<0.0001
Active Ratio	Ascites	0.8617	0.8629	72.7	<0.0001
Active Ratio	CVS	0.8036	1.0074	54.4	<0.0001
Active Ratio	Plasma	0.7828	0.7905	52.0	0.0002
DPP4 (ng mL^−1^)	Ascites	0.5598	0.1124	75.8	0.2139
CA125	Plasma	0.8262	0.6171	200.9	<0.0001

## Data Availability

See Appendix A for individual patient details.

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
