# Peer review of "Active Ratio Test (ART) as a Novel Diagnostic for Ovarian Cancer"

_diagnostics, 2021, doi:10.3390/diagnostics11061048_

Round 1

Reviewer 1 Report

The manuscript is well written and the study is excellent regarding experiment design as well as scientific significance. The Reviewer does not have any meaningful comments to this paper.

Author Response

Thank you very much for your review and feedback.

Reviewer 2 Report

The article presents the results of a fairly serious study. There are several questions / comments to the authors: 1. CXCL10 is an IFNγ-dependent ligand, so it can be differentially expressed in a variety of pathologies (diabetes, rheumatoid arthritis, etc.). Did you take into account the presence of concomitant pathologies that may affect the result of the study? 2. CXCL10 can be detected in a variety of cancers. are there any prospects for its use in diagnostics in an independent version? Or is its use limited to the differential diagnosis between malignant and non-malignant ovarian pathologies? 

Author Response

1. It is well established elevated CXCL10 is present in other pathologies with an inflammatory component, as highlighted by the reviewer. However the contribution of post-translationally processed CXCL10 in these conditions is not clear. One exception to this is in inflammatory liver disease caused by HCV infection, where CXCL10 and plasma DPP4 correlate with severity of liver disease in chronically infected HCV patients (see introduction and refs 16-18 therein).

We agree with the reviewer’s comment, and believe that CXCL10 processing as a mechanism to modulate immune activity is likely have broader implications across a range of diseases. For this reason we chose to investigate the use of cervicovaginal swabs as a sampling method of choice, which eliminates the systemic interference potentially posed by non-gynecological malignancies. Of note, both diabetes and rheumatoid arthritis are common co-morbidities for patients in the age range assessed in this pilot study; however, we have not identified any statistically relevant correlation between ART measurements and the prevalence of these diseases yet.

We are currently also investigating additional samples (non-gynecological cancer types) to expand the scope of confounding groups, to specifically exclude these possibilities. This data will become available at a later date.

2. Elevation of total CXCL10 has been detected in a number of non-ovarian malignancies, and as stated above we are currently assessing multiple cancer types (both gynaecological and non-gynaecological) as potential confounding groups. The contribution of post-translational processing, rather than total CXCL10 (which does not have broad prognostic significance) remains unknown.

It is entirely possible that differentiation between active and inactive CXCL10 may have relevance in other malignancies – for example, as highlighted in the discussion (Barreira da Silva, R., et al.) cleaved CXCL10 has been implicated in colorectal and melanoma models. Specificity of testing for gynaecological cancers was conferred by the use of cervicovaginal swabs, and we are currently exploring this option in a new sample cohort. Ongoing work (beyond this pilot study) will address the question of additional cancer types and overall specificity of the test developed.